# Ionizing Radiation Synthesis of Hydrogel Nanoparticles of Gelatin and Polyethylene Glycol at High Temperature

**DOI:** 10.3390/polym15204128

**Published:** 2023-10-18

**Authors:** Patricia Y. I. Takinami, Nelida L. del Mastro, Aiysha Ashfaq, Mohamad Al-Sheikhly

**Affiliations:** 1Center of Radiation Technology, Institute of Energy and Radiation Research-IPEN/CNEN, Av. Prof. Lineu Prestes, 2242, Cidade Universitária, Sao Paulo 05508-910, Brazil; patyoko@yahoo.com (P.Y.I.T.); nelida@usp.br (N.L.d.M.); 2Department of Chemistry and Biochemistry, University of Maryland College Park, College Park, MD 20742, USA; aiysha@umd.edu; 3Department of Materials Science and Engineering, University of Maryland College Park, College Park, MD 20742, USA

**Keywords:** radiation-induced synthesis of nanohydrogels, inter- and intracrosslinking reactions C-centered radicals, AFFFF, hydrodynamic diameter

## Abstract

Nanohydrogel particles of polyethylene glycol (PEG), gelatin (GEL), and PEG–GEL mixtures (MIXs) were synthesized with a high electron beam and ^60^Co gamma-ray radiation. The relatively novel technique of Asymmetrical Flow Field Flow Fractionation (AF4 or AFFFF) coupled to a Multi-Angle Laser Light Scattering (MALLS) detector was mainly used to determine the hydrodynamic diameter (D_h_) of the radiation-synthesized PEG, GEL, and PEG–GEL nanohydrogel particles. Our approach to achieving nanohydrogel particles is to enhance the intracrosslinking reactions and decrease the intercrosslinking reactions of the C-centered radicals of the PEG and GEL. The intracrosslinking reactions of these free radicals were enhanced via irradiation at temperatures of 77–80 °C and using a high dose rate and pulsed irradiation. The shorter average distance between the C-centered free radicals on the backbone of the thermally collapsed PEG and GEL chain, due to the destruction of hydrogen bonds, enhances the intracrosslinking reactions. It was observed that increasing the dose and dose rate decreased the D_h_. DLS results lined up with AF4 measurements. This study provides researchers with a clean method to produce GEL–PEG hydrogels without the use of toxic reagents. Particle size can be tuned with dose, dose rate, and temperature as demonstrated in this work. This is ideal for medical applications as the use of ionizing radiation eliminates toxicity concerns and provides simultaneous sterilization of the material.

## 1. Introduction

Hydrogels are 3-D polymeric networks that consist of crosslinked polymers that have high water retention properties. Hydrogels derived from gelatin are highly desirable materials used in drug delivery, bio-sensing, wound dressing, tissue engineering, and other medical applications due to their high biocompatibility and biodegradability [1]. The incorporation of polyethylene glycol (PEG) into the structure of gelatin-based (GEL) hydrogels greatly improves their stability. Most methods outlined in the literature around the synthesis of these hydrogels involve the use of toxic initiators, monomers, surfactants, etc. Therefore, these hydrogels are not ideal for biomedical applications and must undergo various treatments to be considered safe for human usage.

Some excellent reagent-free crosslinking techniques include the use of low LET-ionizing radiation such as gamma rays and fast electrons. These techniques do not require the use of chemical crosslinkers, which eliminates the risk of accidental incorporation of toxic byproducts into an otherwise safe material [2,3,4,5,6,7]. Radiation techniques provide an added benefit of simultaneous sterilization.

As gelatin (GEL) is derived from the hydrolysis of collagen (a major component of bones, skin, and tendons), it is unlikely to cause irritation or be incompatible with biological applications. However, the network structure of gelatin is highly unstable as it is primarily composed of hydrogen bonding. The sol–gel transition temperature of gelatin is below 37 °C, and at temperatures above the gel temperature, these networks are easily destroyed. This is less than ideal for medical applications, as the average human body temperature ranges from 36.1 to 37.2 °C. Furthermore, gelatin in a dry state is highly brittle with low flexibility and a fast degradation rate. Thus, increasing crosslinking of the gelatin and incorporating co-polymer additives within the gelatin structure are often induced to improve the gelatin’s thermal and mechanical properties.

Polyethylene glycol (PEG) is often used as a polymer to blend with the gelatin to enhance its mechanical characteristics [8,9,10,11,12,13,14,15,16]. As a Food and Drug Administration (FDA) approved synthetic polymer for use in the human body, it has many potential biological applications. Furthermore, the crosslinking density of PEG and its derivatives can be adjusted with relative ease [17]. For example, hybrid polyethylene glycol diacrylate (PEGDA)/gelatin hydrogels synthesized using an electron accelerator have shown an increase in mechanical strength with rising irradiation dose (and crosslinking density) up to 205% compared with that of pure gelatin hydrogels [18]. Furthermore, PEG and its derivatives are highly hydrophilic and biocompatible with many active sites that can readily be chemically modified.

During the radiation-induced synthesis, the bonds within the polymeric chain are broken (e.g., C–H bonds undergo homolytic cleavage), leading to the formation of radicals. Direct bond cleavage, however, is less common when the polymers are present in an aqueous medium. In this scenario, the radiolysis of water initiates the formation of short-lived reactive radical species that abstract hydrogens from the backbone of the polymer, resulting in radical formation [19]. The radicals on the polymer undergo inter or intra-molecular crosslinking reactions that promote the stability and durability of the structure.

Radiolysis of water leads to the formation of the following oxidizing and reducing species with the following radiation-chemical yields (G-values) in micromole per absorption of one joule (μmol/Joule).
H_2_O → e_aq_^−^, ^•^OH, H, H_2_O_2_, H_2_, H_3_O^+^

G(e_aq_^−^) = G(^•^OH) = G (H_3_O^+^) = 0.29, 
G(H) = 0.08 G(H_2_O_2_) = 0.08   G(H_2_) = 0.04 (1)

To increase the yield of ^•^OH prior to irradiation, aqueous polymer solutions are saturated with N_2_O to convert the eaq^−^ into ^•^OH as follows:N_2_O + e_aq_^−^ + H_2_O → ^•^OH + OH^−^ + N_2_   k = 9.1 × 10^9^ L mol^−1^ s^−1^(2)

It is known that ^•^OH radicals abstract H atoms and add to the double bond of the polymer chain leading to the formation of C-centered radicals. Therefore, under our experimental conditions, the reactions of ^•^OH with GEL and PEG can be outlined as follows:^•^OH + GEL → (GEL(-H))^•^ + H_2_O(3)
^•^OH + PEG → (PEG(-H))^•^ + H_2_O(4)

These carbon-centered radicals formed on the backbone of the polymer chains then undergo inter- and intracrosslinking to form nanoparticles and the 3-D networks of the hydrogels, respectively.

In addition to the intracrosslinking of these free radicals, as a result of the crosslinking reactions of two free radicals on the backbone of one chain, intercrosslinking reactions can take place as follows:2(GEL(-H))^•^ → GEG-GEL(-2H)(5)
2(PEG(-H))^•^ → PEG-PEG(-2H)(6)
(GEL(-H))^•^ + (PEG(-H))^•^ → GEL-PEG(-2H)(7)

We have previously shown that the conformation of the hydrophilic polymers undergoes a collapse process at high temperatures due to the destruction of their hydrogen bonds with water [20,21,22]. Therefore, our strategy to synthesize hydrogel nanoparticles is to irradiate the GEL and PEG aqueous solutions at high temperatures, whereby the chains have collapsed due to the destruction of hydrogen bonds between the polymer chains and water molecules. The distance between the C-centered free radicals formed on the backbone of one collapsed chain is much shorter than the distance between two C-centered radicals on the backbone of different two chains (intercrosslinking). In the case of intercrosslinking, the diffusion coefficient plays a major role. On the contrary, for intra-crosslinking, the diffusion coefficient does not have any effects on the reaction since both free radicals are located on the same molecule.

Prior to irradiation, the GEL and PEG chains have already clustered into microgels through very strong hydrogen bonds with a relatively high hydrodynamic diameter (D_h_). Due to their affinity to water via hydrogen bonds, these molecules undergo swelling processes that increase their D_h_ tremendously. In particular, the water-induced conformation of the strong effect of water on PEG and GEL resides in the properties of H-bond asymmetry in solute–solvent interactions. First, the conformation of the PEG chains is mainly determined by the first solvation shell of the water layer. Secondly, it was shown that in dilute aqueous solutions of PEG, H_2_O molecules form strong H-bonds with two adjacent ethylene glycol oxygen atoms of the same on the backbone of the same chain. Such a bridged H-bonded structure is favored if the adjacent C_i_–C_i+1_ bond attains gauche form and the adjacent C_i_–O_i_ and C_i+1_–O_i+1_ bonds remain in trans conformation [23,24]. Thirdly, it has been reported that PEG in aqueous solutions undergoes coil–helix transition that occurs simultaneously with phase separation of its solutions. At room temperature, the PEG has a coil shape, but at elevated temperatures, PEG chains have a helix shape and undergo phase separation [25].

It has been reported that GEL hydrogels incorporated all available water when irradiated with relatively low doses (up to 25 kGy). At high doses, the GEL hydrogel presented better mechanical properties, but in these cases, where crosslink density is higher, part of the available water would be compressed by crosslinked structure and not incorporated into gelatin hydrogel.

Gamma irradiation entails long exposure times to achieve the desired dose, with dose rates close to 2–7 kGy/h [26]. Electron irradiations, on the other hand, allow for higher dose rates (up to 200–300 kGy/h). Gamma rays can penetrate material deeper than electron beams (EBs) and are the better choice for large bulky materials. However, electron beams offer more flexibility in the penetration depth by changing the electron energies. In the presence of oxygen, the macro radicals will react with the oxygen and not crosslink as readily. Therefore, irradiations must be conducted under inert conditions.

In this work, GEL, PEG, and GEL–PEG (MIX) systems were irradiated at different doses using a pulsed electron beam accelerator and a ^60^Co source. The particle size distributions of the hydrogels were studied using Dynamic Light Scattering (DLS) and Asymmetrical Flow Field Flow Fractionation (AF4 or AFFFF) coupled to a Multi-Angle Laser Light Scattering (MALLS) detector. Atomic force microscopy (AFM) was used to study the surface topography of the gels.

## 2. Materials and Methods

Samples of type B gelatin (bovine, food grade) from Gelita of Brazil (240 Bloom/10 mesh) were used for the development of hydrogels. The average molecular weight M_n_ of GEL is 140,000 g/mol. Polyethylene glycol (PEG) was acquired from Bio World with an M_n_ of 4000 g/mol.

### Sample Preparation

GEL, PEG, and MIX (mixture of GEL and PEG) solutions were prepared in deionized water (DI), Millipore Direct Q system, resistivity of 18.2 MΩ, as follows: 0.01% (*w*/*v*) GEL; 0.1% (*w*/*v*) PEG; and MIX in the ratio (1:10) of GEL 0.01% (*w*/*v*) and PEG 0.1% (*w*/*v*).

The samples were placed in ultrasound for 10 min. They were heated in a water bath at 60 °C for 15 min under constant stirring. Thus, each solution was transferred to 10 mL glass vials with a cap and silicone septum. Then, the N_2_O gas was used to convert the e_aq_^−^ to ^•^OH.

The irradiation conditions were modified by varying the dose rate from 16.5 Gy/3 µs pulse to 70.5 Gy/3 µs pulse, and the total dose used ranged from 1 to 15 kGy. A homogeneous dose distribution on the surface of the sample container was confirmed by the beam mapping technique using radiochromic dosimetry film FWT.

The particles and size characterization of non-irradiated and irradiated GEL, PEG, and MIX solutions was performed using the Dynamic Light Scattering (DLS) technique and Asymmetrical Flow Field Flow Fractionation (AF4) coupled to a Multi-Angle Laser Light Scattering (MALLS) detector (See Figure 1). To perform the DLS tests, 2 mL of the sample was required for each reading. For AF4 analysis, 1 mL of the sample was placed in 1.5 mL vials.

The topographic surface analysis of GEL and MIX samples was performed in a liquid and dry environment. In the liquid environment, 100 μL of the diluted sample solution was deposited on the cleaved mica disk with 9.9 mm diameter PELCO^®^, and for the dry surface test, 10 μL of the sample was deposited and maintained at room temperature until completely dry to analyze.

## 3. Results/Discussion

The radiolytically produced ^•^OH radicals from the radiolysis of H_2_O can abstract H- atoms from the backbone of the GEL and PEG chains, producing C-centered radicals. In addition to the CH_2_ group, the molecular structures of GEL and PEG contain C=O, NH, C=NH_2_, and C-O-C. It is expected that the abstractions of hydrogen will take place at secondary carbon atoms, C-CH_2_-C, since they are present in the GEL backbone at higher numbers relative to the other carbon groups. The abstraction of H-atoms by ^•^OH from the secondary carbon proceeds at reaction rate constants of ~10^8^ mol^−1^s^−1^ [28].

It has been established previously that at elevated temperatures, the polymer chains undergo thermal collapse due to the destruction of hydrogen bonding between the polymer and its aqueous medium [21,22]. Studies have shown that increasing the dose tends to increase the intercrosslinking of high-concentration GEL samples [29,30,31]. However, by increasing the temperature and decreasing the concentration of GEL and PEG, we believed the intracrosslinking will dominate and more compact hydrogels will be synthesized.

In order to investigate the effects of the dose rates on the D_h_ of the collapsed chains of the GEL, PEG, and the mixtures of GEL+PEG in aqueous solutions, we used 16.5 Gy/3 µs and 70.5 Gy/3 µs at 77 °C. We kept the dose repetition at 60 pulses per second.

The changes in hydrodynamic diameter (D_h_) of analyzed samples irradiated at 77 °C in pulsed EB at dose rates of 16.5 Gy/3 µs pulse and 70.5 Gy/3 µs pulse are shown in Figure 2 and Figure 3, respectively. These samples were analyzed with a DLS instrument.

It was observed that the D_h_ of samples decreased with increasing dose and dose rate/pulse. Results were obtained for GEL solutions irradiated at 16.5 Gy/3 µs pulse at 2 kGy (152.4 ± 76.2 nm) and 5 kGy (128.8 ± 36.8 nm), and at 70.5 Gy/pulse the values ranged from 132 ± 23 nm for 2 kGy to 125 ± 19 nm for 5 kGy.

The PEG sample showed similar performance at the same analyzed dose rate for 5 kGy at 16.5 Gy/3 µs pulse (79.3 ± 0.8 nm) and at 70.5 Gy/3 µs pulse (58 ± 2 nm). For MIX, there was a decrease in the D_h_ values at 5 kGy for analyzed dose rates at 16.5 Gy/3 µs pulse (122.8 ± 36.2 nm) and at 70.5 Gy/3 µs pulse (50 ± 3 nm).

The D_h_ results obtained for samples irradiated with a ^60^Co source at 77 °C are shown in Figure 4. The GEL samples irradiated at 70 kGy h^−1^ for which at 2 kGy and 5 kGy presented a D_h_ of 139.8 ± 28.3 nm and 94.9 ± 16.8 nm, respectively. The MIX showed smaller D_h_ compared to the GEL sample for 2 kGy (60.2 ± 6.8 nm) and 5 kGy (40.0 ± 6.1 nm).

Table 1 and Table 2 present the D_h_ hydrogels synthesized at high temperature (77 °C) by ^60^Co and pulsed EB radiation process at a dose rate of 70.5 Gy/pulse analyzed by the AF4 technique.

The GEL samples irradiated with gamma presented a D_h_ of 127 ± 28 nm at 2 kGy and 83 ± 26 nm at 5 kGy. The MIX presented smaller D_h_ than GEL, in the same dose, respectively: 68 ± 25 nm and 35 ± 4 nm.

The hydrogels irradiated with pulsed EB showed lower D_h_ than those irradiated in a gamma source, as the D_h_ values showed at 5 kGy for GEL (84 ± 8 nm) and MIX (26 ± 9 nm).

Furthermore, the use of a high dose/pulse (70.5 Gy/pulse) was shown to be more effective at higher temperatures (77 °C). It has been previously observed that nanogel formation in deoxygenated dilute aqueous solutions of hydrophilic polymers and the use of irradiation at high dose/pulse led to the simultaneous formation of many radicals in each polymer chain, which occurs preferentially by intramolecular recombination [32,33]. At higher temperatures (above 50 °C), the collapse of the GEL and PEG molecules occurs. Thus, the interactions between repeating units in the same polymer chain become more favorable due to the breaking of hydrogen bonds between water-polymers, leading to a more coiled or contracted conformation [34]. This effect may reduce the average distance from the GEL, PEG, and MIX radicals generated within a single chain, and subsequently lead to the formation of more compact nano-hydrogels, which are crosslinked by intramolecular recombination.

A high pulse repetition rate in large time-averaged [^•^OH] generates a larger number of carbon-centered free radicals on the same backbone of the GEL chain than the number of such radicals obtained at a low pulse repetition rate per unit time. This is because increasing the number of carbon radicals coexisting on the same chain per unit of time promotes intramolecular radical recombination rather than the reaction of intermolecular radicals caused by diffusion [21,35].

Attempts were made to determine the molar mass of synthesized GEL, PEG, and MIX hydrogels irradiated with ^60^Co and pulsed EB in different doses at 77 °C. Since the GEL and PEG are already in gel forms and exist in phase separation, no meaningful results were obtained.

Figure 5 shows the MIX solutions at different concentrations after irradiation. Samples were saturated with N_2_O prior to irradiation (to convert e_aq_^−^ to ^•^OH, see Equation (2)), and were initially visually clear solutions without any observed turbidity. The sample solution “A”, with a concentration set at a ratio of 1:10 (GEL: 0.01% (*w*/*v*), PEG: 0.1% (*w*/*v*)), maintained its original clear liquid after irradiation with pulsed EB at 77 °C with a dose of 5 kGy at a dose rate of 75.5 Gy/3 µs.

The “B” sample showed a slight translucency compared to the sample irradiated with gamma rays using the same concentration. The other tested samples showed different behavior with the formation of macroscopic gels separated from the liquid phase due to the different concentrations of samples C (1:1), D (1:4), and D (1:40) of GEL and PEG, respectively. For the irradiation of sample A, the MIX (1:10) sample, irradiated at 5 kGy at 77 °C to 70.5 Gy/3 µs pulse), produced a smaller D_h_ (50 ± 3 nm) than those prepared at 16.5 Gy/3 µs pulse (122.8 ± 36.2 nm) at the same temperature. This behavior can be attributed to the collapsed conformation of the polymer chain, which is produced at high temperatures, and therefore leads to a more densely crosslinked structure formation [21]. It should be mentioned that at room temperature, due to the presence of strong hydrogen bonds, the gels have a very high D_h_ due to swelling. This prevented us from achieving meaningful data on D_h_.

In the present study, Atomic Force Microscopy (AFM) measurements were also used to compare the surface morphology of GEL and MIX nanoparticles in a liquid environment on mica (Figure 6). The control GEL sample (a) showed a mean diameter of 30.5 nm. The GEL (b) and MIX (c) irradiated samples at 15 kGy presented an average diameter of 23.5 nm and 16.5 nm, respectively. The adhesion of the sample to the surface of a solid substrate is essential in order to prevent pullout during the scan. This is a critical problem for organic and biological samples that are fragile and have a weak interaction with most substrates [36].

The electrostatic properties of mica surfaces coated with gelatin were determined and were found to be dominant in surface–surface interactions of aqueous systems. It is known that mica surfaces are negatively charged in pure water [36]. When a thin layer of gelatin is deposited on a mica surface, the loading surface is made up of two contributions: the mica load and the load from gelatin [37].

## 4. Proposed Mechanism Crosslinking of PEG–GEL

Despite the fact that the elevated temperatures increase the translational diffusive motion of the chains in aqueous solutions, they also destroy the hydrogen bonds between the GEL–GEL, GEL–H_2_O, PEG–PEG, PEG–H_2_O, and PEG–GEL in the mixture solutions. This leads to the collapse of the chains and consequently decreases their D_h_. While the increase in the translational diffusive motion of the free radical chains enhances the probability of the collisions and leads to the rise of the intercrosslinking reactions, the collapse of the chains decreases the distance between the free radicals on the backbone of the same chains and consequently enhances the probability of the intracrosslinking reactions.

The PEG, as mentioned earlier, at 77–80 °C, undergoes phase separation of its solution. Accordingly, the two-phase system was irradiated. Therefore, it is expected that, at the upper phase, the water content is high, and the chains of the helix structure are far away from each other. Hence, the free radicals on the backbone of the chains interact with each other via intracrosslinking reactions. On the other hand, at the PEG-rich phase, one would expect that both intercrosslinking and intracrosslinking take place.

The radiolysis of the GEL and PEG mixture in aqueous solutions may lead to the crosslinking between them. At 77–80 °C, where the system is in a phase separation state, the PEG and GEL chains are very close to each other. The high translation diffusion of these chains enhances the collision between GEL and PEG C-centered radicals. It should also be mentioned that the hydrophobic parts of the GEL will be inside the coil, while the hydrophilic part will be outside the coil, facing the aqueous medium. This will facilitate the abstraction reactions of the ^•^OH radicals to produce GEL(-H)^•^. Figure 7 outlines the proposed mechanism of the PEG–GEL crosslinking. It is also expected that in the polymer-rich phase, the PEG–GEL crosslinking reaction competes with PEG–PEG and GEL–GEL intercrosslinking reactions, as well as the intracrosslinking relations of GEL and PEG.

## 5. Conclusions

In conclusion, it seems that the high dose rate and shorter pulses are the major factors to enhance the intracrosslinking reactions in the PEG and GEL aqueous solutions. The PEG phase separation includes a polymer-rich region where the intracrosslinking reactions are enhanced. At a very high dose rate, and under pulsed conditions, it is expected that decay of the free radicals along the backbone of the GEL and PEG follow the dispersive kinetics model. This model is described by Plonka’s model and it assumes that the free radicals with a shorter distance between them along the backbone of the same chain combine much faster than those that are separated by a larger distance [38].

## Figures and Tables

**Figure 1 polymers-15-04128-f001:**
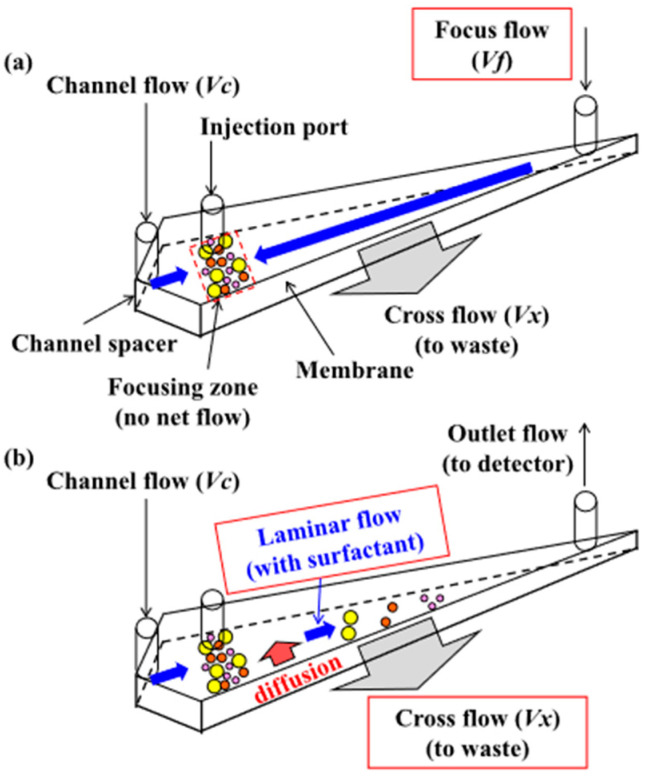
Schematic of the flow profile of the Asymmetrical Flow Field Flow Fractionation (AF4 or AFFFF) instrumentation [27]. (**a**) is the injection phase. (**b**) laminar flow and separation in field applied laminar flow.

**Figure 2 polymers-15-04128-f002:**
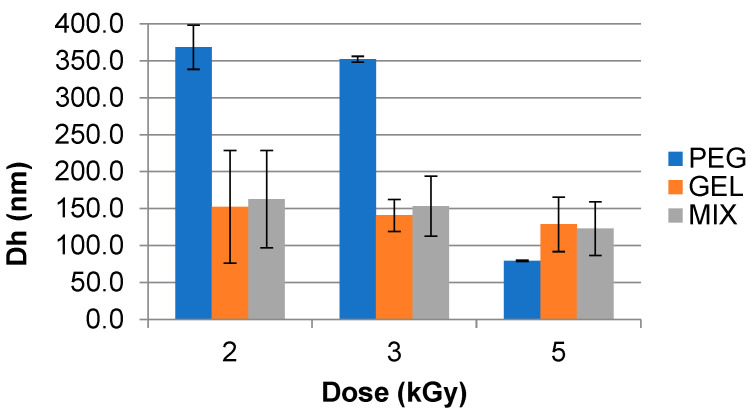
Hydrodynamic diameter (D_h_) of synthesized gelatin (GEL), polyethylene glycol (PEG), and PEG–GEL (MIX) hydrogels irradiated in a pulsed electron beam (EB) at a dose rate of 16.5 Gy/3 µs pulse at 77 °C. Analyzed using Dynamic Light Scattering (DLS) instrumentation.

**Figure 3 polymers-15-04128-f003:**
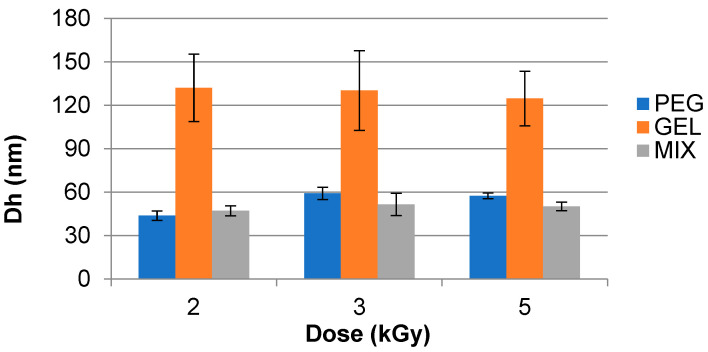
Hydrodynamic diameter (D_h_) of synthesized Gelatin (GEL), polyethylene glycol (PEG), and PEG–GEL (MIX) hydrogels irradiated in a pulsed electron beam (EB) at a dose rate of 70.5 Gy/3 µs pulse at 77 °C. Analyzed using Dynamic Light Scattering (DLS) instrumentation.

**Figure 4 polymers-15-04128-f004:**
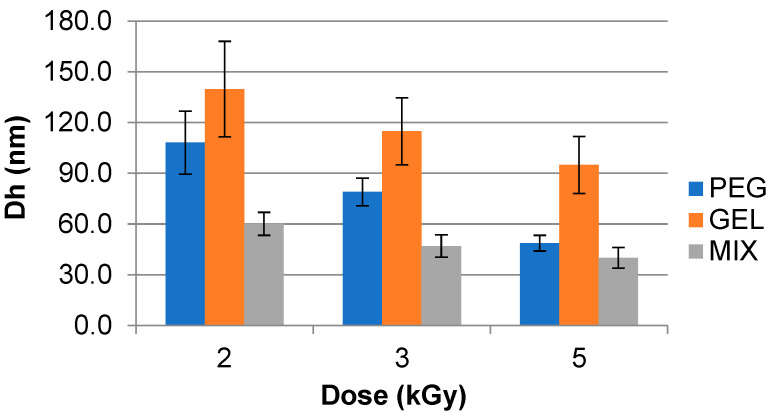
Hydrodynamic diameter (D_h_) of synthesized gelatin (GEL), polyethylene glycol (PEG), and PEG–GEL (MIX) hydrogels irradiated in ^60^Co source at 70 kGy h^−1^ at 77 °C. Analyzed using Dynamic Light Scattering (DLS) instrumentation.

**Figure 5 polymers-15-04128-f005:**
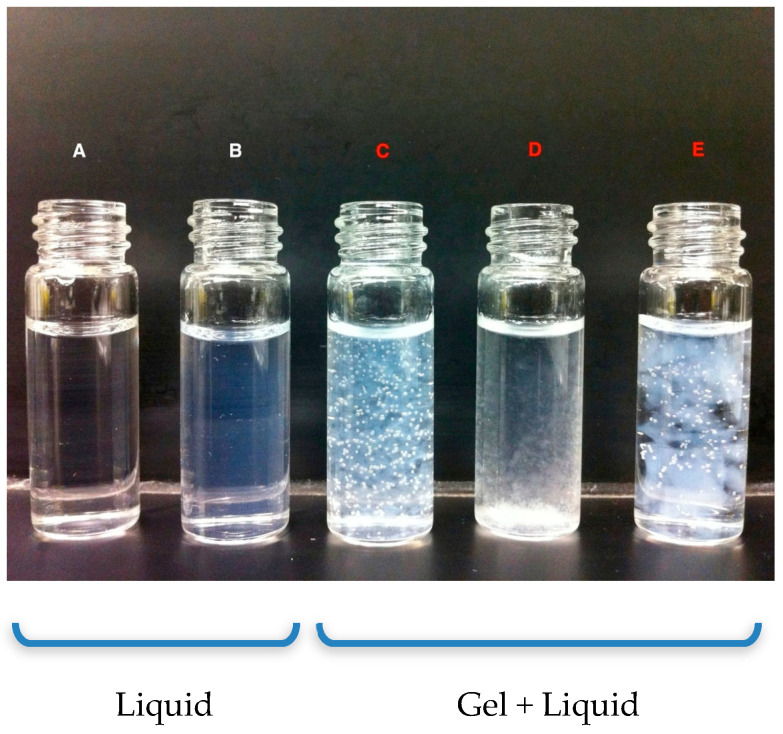
Irradiated samples: sample A synthesized at at 77 °C, irradiated at 15 kGy using pulsed electron beam (EB), and samples B to E at 15 kGy by gamma rays (γ-rays) synthesized at 20 °C in different MIX solutions concentrations.

**Figure 6 polymers-15-04128-f006:**
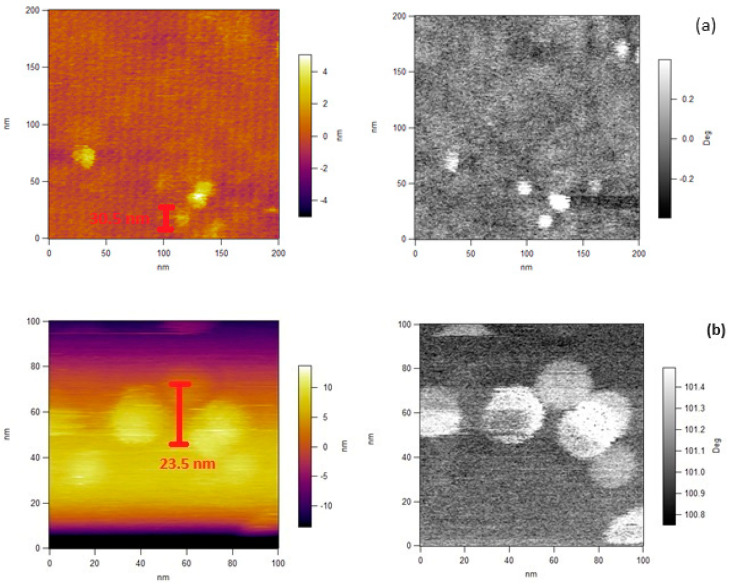
Liquid environment Atomic Force Microscopy (AFM) micrographs of control GEL (**a**) and irradiated GEL at 15 kGy (**b**) hydrogels with pulsed EB and MIX (**c**) hydrogels at 15 kGy.

**Figure 7 polymers-15-04128-f007:**
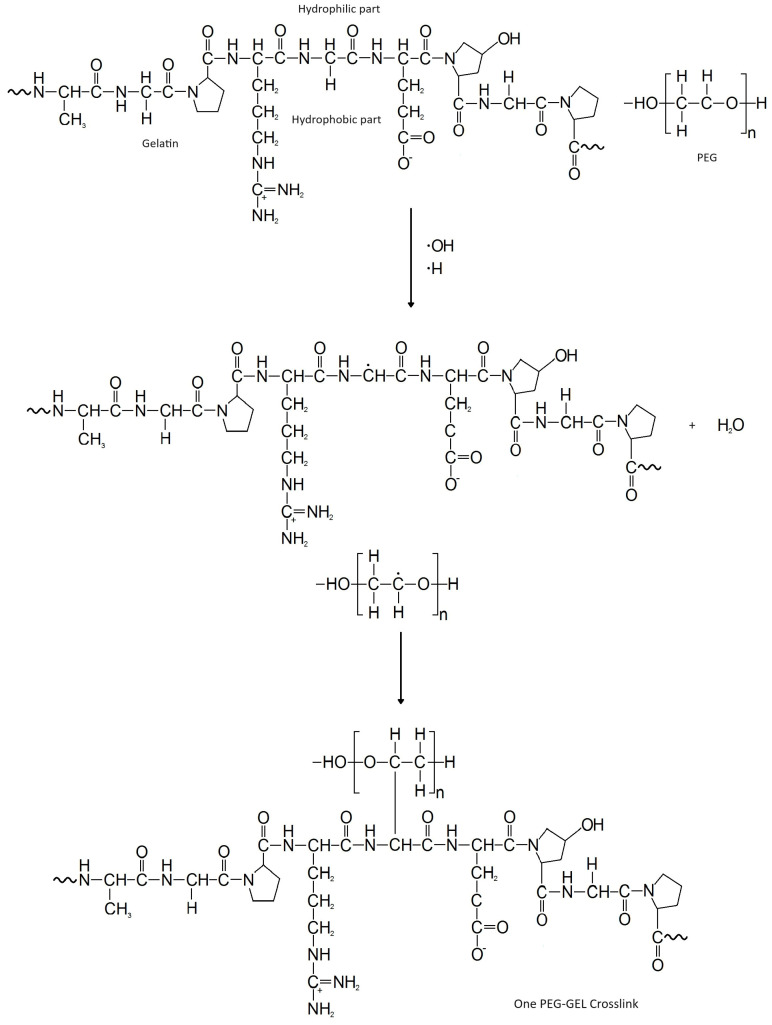
Proposed mechanism for PEG–GEL crosslinking. The hydroxyl radicals and H-atoms are primarily responsible for the abstraction of hydrogens from the backbone of PEG and GEL molecules. One crosslink is shown in the figure, when in reality many take place.

**Table 1 polymers-15-04128-t001:** Asymmetrical Flow Field Flow Fractionation (AF4) hydrodynamic diameter (D_h_) of synthesized GEL, PEG, and MIX hydrogels irradiated in ^60^Co source at 70 kGy h^-1^ in different doses at 77 °C.

Dose	D_h_ (nm)		
(kGy)	GEL	PEG	MIX
2	127 ± 28	73 ± 28	68 ± 25
3	94 ± 17	50 ± 4	48 ± 11
5	83 ± 26	38 ± 7	35 ± 4

**Table 2 polymers-15-04128-t002:** Asymmetrical Flow Field Fractionation (AF4) hydrodynamic diameter (D_h_) of synthesized GEL, PEG, and MIX hydrogels irradiated in a pulsed EB at a dose rate of 70.5 Gy/pulse at 77 °C.

Dose	D_h_ (nm)		
(kGy)	GEL	PEG	MIX
5	84 ± 8	38 ± 25	26 ± 9
10	67 ± 8	27 ± 12	25 ± 10
15	62 ± 48	28 ± 13	30 ± 15

## Data Availability

Not applicable.

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
