# Peer review of "Ionizing Radiation Synthesis of Hydrogel Nanoparticles of Gelatin and Polyethylene Glycol at High Temperature"

_polymers, 2023, doi:10.3390/polym15204128_

Round 1

Reviewer 1 Report

Journal: Polymers

Manuscript ID: polymers-2587213  

Title: Ionizing Radiation Synthesis of Hydrogel Nanoparticles of Gelatin and Polyethylene Glycol at High Temperature.

Dear Authors,

I have completed a thorough assessment of your manuscript and would like to commend you for the well-structured and captivating content. Nevertheless, I do have a number of suggestions that could potentially elevate the caliber of your work.

1. It is important to ensure that the main findings presented in your paper are adequately summarized in the Abstract.

2. Refine the Objective: Highlight the unique aspects that set your study apart from the existing literature on the same topic. Clearly convey the new insights your research brings to readers, and make sure these facets are prominently expressed in both the Objective and Abstract sections, making your objective more intriguing.

3. Abbreviations: Maintain consistent and clear usage of abbreviations throughout the text, including in the Abstract. Provide the full expansion of each abbreviation when it is first used to avoid any potential confusion.

4. Captions for Tables and Figures: Double-check that the captions sufficiently explain the information presented. Additionally, provide a comprehensive explanation of any abbreviations used in the captions to eliminate any uncertainty.

5. While the discussion section holds promise, I recommend that the authors incorporate a comparison with data obtained by other research groups to bolster their findings. Therefore, I suggest revising this section, focusing on a more comprehensive analysis and discussion of the results, while also drawing relevant comparisons with existing research.

6. The Conclusion section is excessively long, and I propose shortening it. Consider incorporating potential implications or discussing how your findings might influence the field based on the considerations outlined in the Conclusions section.

Author Response

We would like to thank the reviewer for their invaluable comments. Here are our responses to the points made:

  1. It is important to ensure that the main findings presented in your paper are adequately summarized in the Abstract.

We agree. We have changed the language of the abstract to provide a general overview of the paper and findings.

  1. Refine the Objective: Highlight the unique aspects that set your study apart from the existing literature on the same topic. Clearly convey the new insights your research brings to readers, and make sure these facets are prominently expressed in both the Objective and Abstract sections, making your objective more intriguing.

We agree and thank you for this comment.  We have added the following sentences:

“This study provides researchers with a clean method to produce GEL-PEG hydrogels without the use of toxic reagents. Particle size can be tuned with dose, dose rate and temperature as demonstrated in this work. This is ideal for medical applications as the use of ionizing radiation eliminates toxicity concerns and provides simultaneous sterilization of the material.”

  1. Abbreviations: Maintain consistent and clear usage of abbreviations throughout the text, including in the abstract. Provide the full expansion of each abbreviation when it is first used to avoid any potential confusion

Thank you for pointing this out. We have updated the abbreviations to make sure that the full expansion is provided first.

  1. Captions for Tables and Figures: Double-check that the captions sufficiently explain the information presented. Additionally, provide a comprehensive explanation of any abbreviations used in the captions to eliminate any uncertainty.

We have updated the captions to sufficiently explain the information being presented. We also expanded on any abbreviations in the caption to eliminate uncertainty.

  1. While the discussion section holds promise, I recommend that the authors incorporate a comparison with data obtained by other research groups to bolster their findings. Therefore, I suggest revising this section, focusing on a more comprehensive analysis and discussion of the results, while also drawing relevant comparisons with existing research.

We could not find papers that looked at particle size comparisons for PEG-GEL hydrogel composites synthesized using e-beam vs gamma source. However, we agree with this comment and made the following changes.

  • Referenced previous studies that show the thermal collapse of polymers at elevated temperatures
  • Included some papers on the individual crosslinking of GEL [more papers were referenced in the introduction]
  • Changed the language to be more concise and clear
  • Moved phase separation discussion from the conclusion to the discussion section
  1. The Conclusion section is excessively long, and I propose shortening it. Consider incorporating potential implications or discussing how your findings might influence the field based on the considerations outlined in the conclusions section.

We have significantly shortened the conclusion and changed the language so that it is more concise.  

Thank you again for your feedback!

Reviewer 2 Report

In this research article authors have described the gamma radiation and high electron beam method to synthesize the crosslinked hydrogel. Authors has described the changes in the particle size and AFM microscopic images to show the successful crosslinking. However, using gamma radiation-based crosslinking of hydrogel is well reported and reviewer finds difficult to find the novelty and significance of this research article.  My comments are as follows; 

  1. Since authors are claiming that this formulation is non-toxic, so they must provide some cell toxicity data of this polymer. Since this reaction generated free radicals so it important to investigate the effect.  

  1. Chemical characterization of hydrogel should be performed, DLS analysis in not just enough to justify the work.  

  1. Water retention capacity of hydrogel should be supplemented.  

  1. Morphological characterization also should be performed to enhance the research quality. 

ok

Author Response

We would like to thank the reviewer for their feedback. To address the points raised:

  1. Since authors are claiming that this formulation is non-toxic, so they must provide some cell toxicity data of this polymer. Since this reaction generated free radicals so it important to investigate the effect.

We have cited references in the introduction that cite articles where GEL and PEG polymeric materials are used for biomedical applications. All radicals are very short lived. We mention this in the manuscript.

  1. Chemical characterization of hydrogel should be performed, DLS analysis in not just enough to justify the work. 
  2. Water retention capacity of hydrogel should be supplemented. 
  3. Morphological characterization also should be performed to enhance the research quality

We agree with these statements. However, we are primarily looking into the induction of crosslinking when we use ionizing radiation. More specifically, we are trying to set experiment parameters such that intra-crosslinking dominates inter-crosslinking hence the importance of particle size. We buy the PEG and GEL from manufacturers therefore we are not actually synthesizing these polymers. We agree that a more comprehensive study would present more data but we believe for the purpose of this article, what we have should suffice.

We thank reviewer again for their feedback.

Reviewer 3 Report

This manuscript is devoted to ionizing radiation synthesis of hydrogel Nanoparticles. The authors described in details the process of synthesis of nanoparticles from gelatine and poly(ethylene) glycol. The article is well written and easy to read and understand. The data are illustrated by 6 figures. The results are well described. However, I have some remarks.

1) It is unclear, how the particles obtained were purified and divided from initial mixtures?

2) instead of Dh at all figures, please, provide the size distribution figures. It allows to understand the formation of nanoparticles and possible aggregates.

3) Please, divide results and discussion parts and add the enhanced discussion on results with literature references.

Author Response

We would like to thank the reviewer for their invaluable feedback. Here is how we have addressed the comments:

  1. It is unclear, how the particles obtained were purified and divided from initial mixtures?

The irradiations are all performed in a water medium and the final products do not need to be purified.

  1. Instead of Dh at all figures, please, provide the size distribution figures. It allows to understand the formation of nanoparticles and possible aggregates.

We thank you for this comment. While we agree wholeheartedly, we do not have the size distributions readily available. However, we would like to point out that the error bars show the range of the size range of the particles, with the average being the reported Dh. We hope the results as reported should suffice.

  1. Please, divide results and discussion parts and add the enhanced discussion on results with literature references.

The format for the journal has the results and discussion section grouped together. However we agree with this comment and have made the following changes to the discussion section to further enhance it

  • Referenced previous studies that show the thermal collapse of polymers at elevated temperatures
  • Included some papers on the individual crosslinking of GEL [more papers were referenced in the introduction]
  • Changed the language to be more concise and clear
  • Moved phase separation discussion from the conclusion to the discussion section

Round 2

Reviewer 2 Report

Author did not provide novelty of of this this work as well as additiional experiments. This research  can not be accepted in present form. 

ok

Author Response

Thank you for your review. We leave the decision with the editors.

Reviewer 3 Report

I think the revised version can be accepted

Author Response

Thank you for your feedback.